# Loss of Rose Fragrance under Chilling Stress Is Associated with Changes in DNA Methylation and Volatile Biosynthesis

**DOI:** 10.3390/genes14030692

**Published:** 2023-03-10

**Authors:** Limei Xie, Xue Bai, Hao Zhang, Xianqin Qiu, Hongying Jian, Qigang Wang, Huichun Wang, Dedang Feng, Kaixue Tang, Huijun Yan

**Affiliations:** 1Flower Research Institute of Yunnan Academy of Agricultural Sciences, Kunming 650205, China; 2Institute of Resource Plants, Yunnan University, Kunming 650000, China; 3National Engineering Research Center for Ornamental Horticulture, Kunming 650000, China

**Keywords:** rose, flower scent, DNA methylation, chilling stress, gene expression

## Abstract

Rose plants are widely cultivated as cut flowers worldwide and have economic value as sources of natural fragrance and flavoring. *Rosa* ‘Crimson Glory’, whose petals have a pleasant fragrance, is one of the most important cultivars of edible rose plants. Flower storage at low-temperature is widely applied in production to maintain quality; however, chilling results in a decrease in aromatic volatiles. To determine the molecular basis underlying the changes in aromatic volatile emissions, we investigated the changes in volatile compounds, DNA methylation patterns, and patterns of the transcriptome in response to chilling temperature. The results demonstrated that chilling roses substantially reduced aromatic volatile emissions. We found that these reductions were correlated with the changes in the methylation status of the promoters and genic regions of the genes involved in volatile biosynthesis. These changes mainly occurred for CHH (H = A, T, or C) which accounted for 51% of the total methylation. Furthermore, transcript levels of scent-related gene Germacrene D synthase (*RhGDS*), Nudix hydrolase 1 (*RhNUDX1*), and Phenylacetaldehyde reductase (*RhPAR*) of roses were strikingly depressed after 24 h at low-temperature and remained low-level after 24 h of recovery at 20 °C. Overall, our findings indicated that epigenetic regulation plays an important role in the chilling tolerance of roses and lays a foundation for practical significance in the production of edible roses.

## 1. Introduction

Rose (*Rosa* L.) plants are among the most important ornamental flowers and have been cultivated for their flowers and fragrance, serving as sources of essential oil [1,2]. At present, more than 40,000 complex hybrid-rose cultivars have been selected through hybrid breeding [3]. In recent years, approximately 20 edible cultivars have been exploited in China [4]. Owing to their beautiful colors, sweet fragrance, and high antioxidant value, edible rose plants have become popular among ornamental plants [5]. Some species and cultivars are natural sources of essential oil for the fragrance industry and are also used as raw materials for edible flowers [6,7]. Yunnan Province is well known for its edible rose plants in China, with more than 4500 hectares under cultivation and an annual production of more than two billion CYN in 2022 [4]. *R.* ‘Crimson Glory’ is one of the most important cultivars of edible rose plants, called ‘the king of edible flowers’ [8]. However, when fresh flowers of ‘Crimson Glory’ are harvested and stored under chilling conditions, their subsequent fragrances are reduced compared with that of fresh flowers. At present, there are few studies on the effects of temperature on floral volatile emissions during chilling storage and the underlying molecular mechanisms.

Floral volatile organic compounds (VOCs) are important components that make up floral scents, including representatives of many functional groups such as terpenes, alcohols, esters, ketones, phenols, hydrocarbons, and aromatic hydrocarbons [9] and play a crucial role in influencing human appetite and consumer preference [10]. The presence or absence of certain compounds can cause large differences in flavor and aroma among species [11]. Abiotic stress factors, such as temperature [12], light intensity [13], and water and salt stress [14], may increase or decrease VOCs biosynthesis and emissions [15]. Terpene VOCs have been proven to improve thermotolerance, helping plants recover rapidly from high-temperature exposure [16]. *Trifolium repens* was shown to emit 58% more floral scent at 20 °C than at 10 °C [17]. There were 17 identified chemical compounds emitted from *Lilium* ‘Siberia’ at 10 °C and 44 at 30 °C [18]. Floral emissions (and thus the amount of volatile compounds) increase with increasing temperature up to a certain point but then decrease thereafter in many plants’ species, for example, *Erica multiflora* L., *Quercus ilex* L., and *Spartium junceum* L [19]. Furthermore, chilling stress is known to significantly reduce volatile content, especially since 12 volatiles derived from amino acids and lipids were influenced in chilling storage [6].

The reduction of VOCs is usually correlated with significantly decreased transcript levels encoding VOCs biosynthesis-related enzymes. Epigenetic factors that affect both genes and transcription factors, including DNA methylation and histone modification, regulate VOCs emissions [20]. In tomatoes, transcriptional changes under low temperatures and flavor loss have been linked to changes in DNA methylation [6]. In sugar beet, DNA methylation levels were found to be significantly reduced by chilling exposure, possibly due to DNA demethylation via the ROS1 pathway [21]. In tea plants, DNA methyltransferase-encoding genes were shown to be downregulated under chilling stress, but demethylase-encoding genes were upregulated [22]. In roses, low temperatures cause DNA hypermethylation within the *RhAG* promoter, which leads to a large increase in the petal numbers of rose plants by promoting stamen petaloidy [23]. Plants have intricate regulatory networks of genes that govern the accumulation of metabolites, and cold acclimation involving physiological, biochemical, and molecular changes. The buds with contrasting chilling requirements are mainly different in redox reactions rather than carbohydrate metabolism [24,25]. Methylation alters the physicochemical characteristics of DNA, resulting in gene silencing and VOCs emission differences, allowing plants to survive and adapt to changes in the environment [26,27].

In this study, to explore the molecular mechanisms underlying the decrease in VOCs emissions under chilling induction, the edible rose ‘Crimson Glory’ was subjected to 6 °C for 24 h, followed by a recovery stage of 24 h at 20 °C. Plants were also grown under 25 °C as a control. The volatile emissions of the plants were compared, and whole-genome bisulfite sequencing (WGBS), as well as RNA sequencing (RNA-seq), were performed. We found that the reduction of volatile compounds is associated with the methylation status of promoters and genic regions for volatile biosynthesis-related enzymes. These results provide insight into the molecular mechanism underlying the decrease in VOCs emissions caused by low temperatures and lay a foundation for the production of edible roses.

## 2. Materials and Methods

### 2.1. Plant Materials

*R.* ‘Crimson Glory’ (Figure 1) was grown at the rose germplasm garden of the Flower Research Institute, Yunnan Agriculture Academic Science, Kunming, China (25.08° N, 102.45° E). ‘Crimson Glory’ at the same floral developmental stages (flowers start to open) were stored under controlled conditions (with 75% relative humidity) and classified into three groups: (i) those stored at 6 °C for 24 h (abbreviated as CGC); (ii) those stored at 6 °C for 24 h then moved to 20 °C for 24 h of recovery (CGR); and (iii) those planted under normal conditions (25 °C) as controls (CGN). Fresh petals were removed, wrapped in aluminum foil, immediately frozen in liquid nitrogen, and stored at −80 °C for WGBS and RNA-seq. Other samples were collected and placed into an icebox for VOCs analysis.

### 2.2. Chilling-Induced Influences on the Volatile Components of R. ‘Crimson Glory’

Volatiles were collected for 45 min from the petals at the opening stage by solid phase microextraction (SPME) with a 75 μm CAR/PDMS SPME fiber (Supelco, Bellefonte, PA, USA). The middle petals of each flower (1 g in total) were added to a headspace bottle. Ethyl decanoate (3 μL) was diluted to 1‰ with hexane and added as an internal standard. The collection bottle of VOCs was stored at a 25 °C thermostat for headspace extraction for 40 min. After extraction, the fiber was immediately inserted into the GC injection port for thermal desorption analysis at 250 °C for 1 min. Three independent replicates were included for each sample [28]. Volatiles were processed by principal component analysis (PCA).

Quantitative analysis was performed by counting peak areas of VOCs with that of the internal standard. The emission amount of each volatile component (ng·g^−1^) = {peak area of compound/peak area of internal standard × concentration of internal standard (ng·μL^−1^) × volume of internal standard (μL)}/Sample Weight (g) [29].

### 2.3. DNA Isolation, Library Construction, and WGBS-Seq

Genomic DNA was acquired using a Tiangen Plant DNA Extraction Kit (DP360), and the quality was detected via gel electrophoresis and spectrophotometrically. The bisulfite convention was performed using the Accel-NGS Methyl-Seq DNA Library Kit (Swift, MI, USA). Converted DNA fragments were amplified via PCR for subsequent sequencing on an Illumina HiSeq 4000 platform (Frasergen, Wuhan, China). The resulting clean read sequences were mapped to the *R. chinensis* ‘Old Blush’ genome sequence (https://lipm-browsers.toulouse.inra.fr/pub/RchiOBHm-V2/, accessed on 7 March 2023) [30]. All raw data of WGBS-seq have been deposited in the NCBI BioProject database (PRJNA902902).

### 2.4. RNA-Seq and Differentially Expressed Gene (DEG) Identification

High-quality RNA was extracted using an EASY Spin Plus Complex Plant RNA Kit, following the manufacturer’s instructions (Genenode Biotech Co., Ltd., Beijing, China). The construction of the libraries and RNA-seq were performed on an Illumina NovaSeq platform in conjunction with paired-ended 150 bp sequences (Wuhan, China). DEGs were selected with fold-change >3 and a *p* < 0.001 as the threshold using the statistical R package. Each treatment was performed in three biological replicates. All raw data were deposited in the NCBI BioProject database under the accession number PRJNA902902.

### 2.5. Identification of Differentially Methylated Regions (DMRs) and DMR-Related Genes

The differentially methylated genomic regions of *R.* ‘Crimson Glory’ in response to chilling stress were identified via the METHYLKIT package [31]. Bins of regions that showed percent methylation differences that were larger than 25% and for which the corrected *p* was ≤ 0.01 were considered to contain DMRs [32]. The DMR-associated genes were identified regions that overlapped with DMRs within the gene body or 2 kb flanking regions in the reference genome, and their sequences were extracted using an in-house Perl script [33]. The regions overlapping between DEGs and DMR-associated genes constructed the function enrichment classification [22].

### 2.6. Gene Validation and Expression Analyses

The sequences of the genes used for qRT-PCR validation are provided in Appendix A. The *RhGAPDH1* gene was used as a reference in the experiments. All the reactions were performed in triplicate, and three different RNA samples that differed from those used to construct the cDNA libraries for the RNA-seq experiments were used. Expression fold-changes were calculated employing the cycle threshold (CT) method [34].

## 3. Results

### 3.1. Chilling-Induced Influences on VOCs Emissions

Low temperatures are known to substantially decrease the contents of volatiles associated with the aroma. To elucidate the mechanisms of VOCs reduction under chilling treatment, we compared the volatiles among *R.* ‘crimson Glory’ treated at 6 °C for 24 h (CGC), ‘Crimson Glory’ grown at 6 °C for 24 h and then moved to 20 °C for 24 h of recovery (CGR), and ‘Crimson Glory’ grown under normal conditions (25 °C) as controls (CGN). A total of 60, 69, and 67 VOCs were identified from CGC, CGN, and CGR, respectively. The results showed significant differences in the total emissions of VOCs among the three treatments (Appendix A). The greatest volatile contents were 31741.29 μg·g^−1^ in CGN and 8101.56 μg·g^−1^ in CGC. Although the increase of VOCs emissions was observed in 24 h recovery at 20 °C volatile content, 28803.08 μg·g^−1^ in CGR was still lower than at CGN (Appendix A).

Twelve main volatile compounds were identified for further analysis (Appendix A). Based on the volatile relative contents, volatiles were distinctly separated by principal component analysis (PCA) under the different treatments, and three biological replicates were found to cluster together (Figure 2B), indicating the data were reliable for subsequent analysis. Volatiles in flowers chilled for 24 h decreased by 65% relative to those of the controls (Figure 2C). Although the 24 h recovery period at 20 °C resulted in increased volatile emissions, the volatile content was still lower than that under normal conditions (25 °C). Most compounds decreased in abundance, and the abundance of terpenoids, alcohols, and esters sharply decreased to at least half of that of the controls at 25 °C (Figure 2C). The volatiles whose abundance sharply decreased after the chilling treatment included citronellol, geraniol, phenylethyl alcohol, benzyl alcohol, benzyl acetate, and hexyl acetate, which together provide a fruit-like fragrance and are responsible for various floral scents. Geranyl acetate was undetectable in CGC. However, the amounts of 1,3,5-trimethoxybenzene (TMB) emitted did not differ considerably compared to these compounds above; on the contrary, showed increasingly in CGR (Figure 2A).

### 3.2. Characteristics of DNA Methylation in R. ‘Crimson Glory’

Epigenetic modification is associated with stress environment, including drought, salinity, heat, and chilling in plants [35]. In this study, we identified potential changes in the DNA methylation involved in chilling stress. WGBS was performed on the samples to assess the occurrence of cytosine methylation. After bisulfite DNA methylation sequencing, approximately 22 Gb of high-quality sequencing data were generated, and the average sequencing depth was approximately 30× per sample (Appendix A). Global methylation analysis of *R.* ‘Crimson Glory’ across the three treatment groups occupied 12.59% in CG, 10.15% in CHG, and 4.13% in CHH (Figure 3A,B). The distribution landscapes of CG, CHG, and CHH methylation were approximately identical in the *R.* ‘Crimson Glory’ genome during chilling treatment. The methylation was mainly exhibited in CHH, which constituted 51% of the total methylated cytosines, while lower proportions of total methylated cytosines were found in CG (24%) and CHG (25%). In response to chilling stress (6 °C) for 24 h, the methylation in CHH increased relative to that in CGN (Figure 3C). Furthermore, we inspected the distribution of CG, CHG, and CHH methylation in genes and their flanking regions within 2 kb. The results found a significant reduction of methylated DNA around the transcription start site (TSS) and the transcription terminal site (TTS), and greater levels in upstream and downstream regions. The CG methylation showed a steady decline of levels from the flank towards TSS regions. The methylations of CHG and CHH were at higher levels in the flanking regions than that in the gene-body regions (Figure 4A).

### 3.3. Identification of DNA Methylation and DEGs

To explore the correction between DNA methylation and different gene expression in rose, we performed the transcriptome analysis of the identical samples with methylation analysis and compared the DEGs and DMRs in rose in response to chilling stress. The results yielded 5008 unique DEGs and 13011 DMRs among three treatment samples (Figure 4B). Of these DMRs, 2391, 3697, and 18534 were separated to CG, CHG, and CHH differences, and the total number of hypermethylated DMRs (34,018) was greater than the total number of hypomethylated regions (14,968), particularly in CHH (Figure 4C).

Sequentially, we found a greater number of downregulated DEGs (3223 unique ones) than upregulated DEGs (1783) under chilling treatment (Appendix A), which probably resulted in a large number of hypermethylated DMRs. Additionally, a total of 24,624 DMR-associated genes were detected, and 4206 genes overlapped with the DEGs. Remarkably, we identified 12 hypermethylated CHH_DMRs and 26 hypomethylated CHH_DMRs among the upregulated DEGs, and 4 hypermethylated CHH_DMRs and 11 hypomethylated CHH_DMRs among the downregulated DEGs (Figure 5B).

The DEG- and DMR-associated gene regions that overlapped were grouped into MapMan BINs [36]. The percentage of upregulated and downregulated genes in the three groups is shown in Figure 5A. Clustering analysis of the different gene sets was performed to group genes with similar expression patterns. These genes had common functions or participated in common metabolic and signaling pathways. Genes whose functions were involved in major monoterpenoid biosynthesis and photosynthesis were strikingly downregulated during chilling stress (Figure 5A). Similar suppression was also found for genes associated with the tricarboxylic acid (TCA) cycle, amino acid metabolism, and secondary metabolism. Moreover, the functional categories of genes that were highly enriched in rose during chilling stress included RNA and protein metabolism.

### 3.4. Identification of DEGs and DMRs Associated with Volatile Biosynthesis

In some plant species, the expression of secondary metabolite-related genes is regulated through the alteration of DNA methylation in response to chilling temperatures [37]. In this study, we observed that several functionally-verified genes, namely, the key methyltransferase gene PMT13 (gene ID: LOC112180472), the cold-responsive (COR) gene COR413PM1 (gene ID: LOC112171899) [38], and cold shock domain-containing protein 3 (CSDE1) (gene ID: LOC112183352). The expression levels of these genes were highly upregulated under chilling stress via RT-qPCR (Figure 6A–C). However, the histone H2B gene (gene ID: LOC112166341) and aquaporin PIP2-7 gene (gene ID: LOC112173307) were significantly downregulated during the chilling response (Figure 6D,E).

We next evaluated the expression of genes encoding enzymes associated with the biosynthesis of volatiles whose abundance significantly changed under chilling treatment in rose, and opposite patterns were observed between transcripts and methylation levels. Germacrene D synthase (*RhGDS*) [39] and Nudix hydrolase 1 (*RhNUDX1*) [40] are involved in the formation of geraniol, and Phenylacetaldehyde reductase (*RhPAR*) [41] levels were much lower after 24 h under chilling and kept low after 24 h of recovery at 20 °C (Figure 6F,G), the patterns of which were similar for the volatile contents (Figure 2). The Phenylacetaldehyde synthase (*RhPAAS*) [2], Phenylacetaldehyde reductase (*RhPAR*), Alcohol acetyltransferase (*RhAAT1*), and Nerolidol synthase (*RhNES*) [42] transcript levels showed strikingly reductions under chilling and did not reach to the normal expression levels after the 24 h recovery period (Figure 6H–L). Orcinol O-methyltransferase (*RhOOMT2*) is involved in catalyzing reactions of DMT biosynthesis [43]. The mRNA of *RhOOMT2* was remarkably lower in the chilling treatment but partially recovered after recovery at 20 °C (Figure 6J).

We further investigated the methylation changes of *RhCOR413*, *RhCSDE1*, *RhNUDX1*, *RhGDS*, and *RhNES* genes in their flanking and genic regions. To some extent, the results showed a different pattern of methylation in gene-body regions and their promoter (Figure 7). For instance, our results showed that the overall methylation levels of *RhPMT13*, *RhCOR413,* and *RhCSDE1* exhibited reduction under chilling treatment, typically for CG methylation of the gene-body regions (Figure 7). Further, increased CHH methylation between the control and chilling stress treatments was also discovered in the promoter and genic regions of scent-related genes *RhNUDX1*, *RhGDS*, and *RhNES* (Figure 7).

## 4. Discussion

Rose scent is one of the most important characteristics used as a source of essential oil for the perfume industry [2]. Volatile emissions, however, are influenced by various biotic and abiotic factors that can lead to substantial changes in emission content and components [19]. Several environmental factors can affect the emission of VOCs from plant tissues, and the effects of temperature on terpene emissions were studied [44,45]. In this study, we found that VOCs emissions significantly decreased in response to chilling. At 6 °C, the total amounts of volatile compounds decreased by 3.6- and 3.9-fold compared to those in the plants subjected to the recovery and normal temperatures, respectively. Similarly, increases in floral emissions were found when *Lilium* ‘Siberia’ plants were transferred from 10 °C to 30 °C [18] and when *Osmanthus fragrans* plants were transferred from 12 °C to 19 °C [46]. The total emissions were found to peak at 30 °C in *Petunia axillaris* [12]. For tomatoes, low temperatures were shown to significantly inhibit the production of volatiles [47]. Plants alter their physiology to optimize floral-scent emissions under specific environmental conditions. Floral physiology governs the production of each compound through the regulation of transcription, gene expression, and biosynthesis of related enzymes, all of which are regulated in physiological adaptations of the biosynthetic or emission mechanisms in flowers [48].

DNA methylation is an important pathway of epigenetic modifications in the regulation of gene expression levels during stress defense in plants [6]. Here, we constructed the DNA methylation map of rose plants’ response to chilling treatment through WGBS-seq. The average methylated levels of the rose genome in CG, CHG, and CHH sequence contexts are 12.59%, 10.15%, and 4.13%, respectively. The results are lower than the overall average methylation levels compared to other plant species. Approximately 21%, 23%, and 56% of the cytosines are methylated at the same sites as those of GC, CHG, and CHH in the tea plant, respectively [22]. In *Bistorta amplexicaulis*, methylated cytosines mainly occurred at the same sites as those of CG and CHG and accounted for 53%, 37%, and only 10% of the total methylated cytosines in the context of CHH [49]. Intriguingly, we found that *R.* ‘Crimson Glory’ has higher CHH methylation levels than other plant species, whose methylation ranged from approximately 1.1% in *Vitis vinifera*, 3% in *B. amplexicaulis*, and 3.2% in *Phyllostachys edulis* [50]. One potential explanation for these discrepancies is the difference in genome size of the different plant species. The correlations between genome size and methylation level have found that plants with larger genome sizes usually possess high levels of DNA methylation [22].

Importantly, we identified some important DMR-associated genes and DEGs in response to the chilling stress of the rose plant. We identified hypomethylated genes with DMRs encoding Phenylalanine ammonia-lyase (PAL) and prenyltransferase (Figure 5A). These proteins are involved in secondary metabolism and were significantly suppressed [36]. A previous study on tomato fruit demonstrated that the metabolic process of terpenoids exhibited dynamic changes during chilling [6]. In addition, we found that a large amount of energy metabolism from the TCA cycle is suppressed during chilling storage. Studies in maize and tomato showed that energy metabolism decreased in response to chilling stress [6,51].

Remarkably, PMT13 and *RhCOR413* were induced in response to chilling stress in rose. A previous investigation showed that AtCOR413pm1 was also induced in response to cold [52]. COR413PM1 is a stress-inducible gene known to be upregulated in response to chilling and drought [53,54]. Furthermore, the aquaporin-encoding gene PIP2-7 was significantly downregulated under chilling treatment, and its expression strikingly decreased during the recovery period. Similarity studies have shown that most PIP genes are downregulated in response to low temperatures in rice [55], maize [56], and Arabidopsis [57]. Aquaporins play a very important role in cellular and molecular mechanisms, driving plant protection against low-temperature stress [58].

In this study, a significant decrease in emissions of VOCs was found for roses stored at 6 °C for 24 h. Despite the 24 h recovery period at 20 °C, the abundance of volatiles was lower than that of the control flowers (Figure 2). This decrease in abundance was correlated with significantly lower transcript levels of genes, especially terpenoid synthases *RhNUDX1*, *RhGDS*, and *RhPAR*. Their products are essential for the synthesis of these volatiles and are associated with major changes in DNA methylation during chilling stress. We found that chilling-caused scent attenuation focused on the metabolic process of PAL, terpenoids, TCA-related energy, and their biosynthesis-related genes [6]. Although the expression of these genes increased after being transferred to 20 ºC for 24 h, the expression of most remained strikingly lower than the expression of the same genes in the control flowers (Figure 6). The results indicated that the expression of genes coding for proteins involved in volatile compound synthesis is sensitive to temperature shifts. The results are consistent with findings reported for tea plants [22], rice [59], maize [52], and papaya [60].

Taken together, these results indicated that the effects of chilling stress in rose might be alleviated by upregulating key COR, PMT13, and PIP genes, and reduced levels of transcripts for some key volatile biosynthesis-related enzymes by increasing their methylation levels in response to chilling. The large number of DMRs and DEGs identified in this study provided an invaluable source for understanding fragrance loss in roses.

## 5. Conclusions

We have demonstrated that roses under chilling lead to significantly reduced emissions of volatiles. This reduction is associated with decreased levels of transcripts for some key volatile biosynthesis-related enzymes involved in the response to chilling. These reduced levels, in turn, are correlated with major changes in the methylation status of promoters and genic regions, suggesting that epigenetic regulation plays an important role in the establishment of chilling tolerance for rose.

## Figures and Tables

**Figure 1 genes-14-00692-f001:**
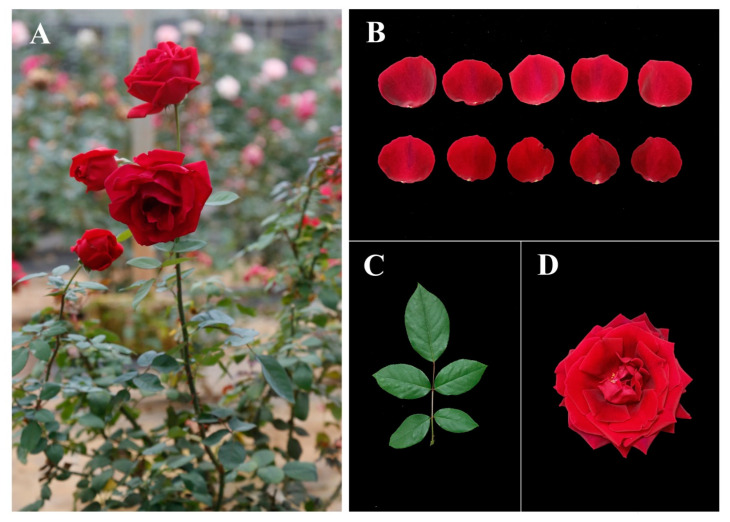
Phenotypes of flowers of *R.* ‘Crimson Glory’. (**A**) Side view of the rose under growing conditions. (**B**) Phenotype of rose flower petals. (**C**) Phenotype of rose leaf. (**D**) Top view of rose flowers at the blooming stage.

**Figure 2 genes-14-00692-f002:**
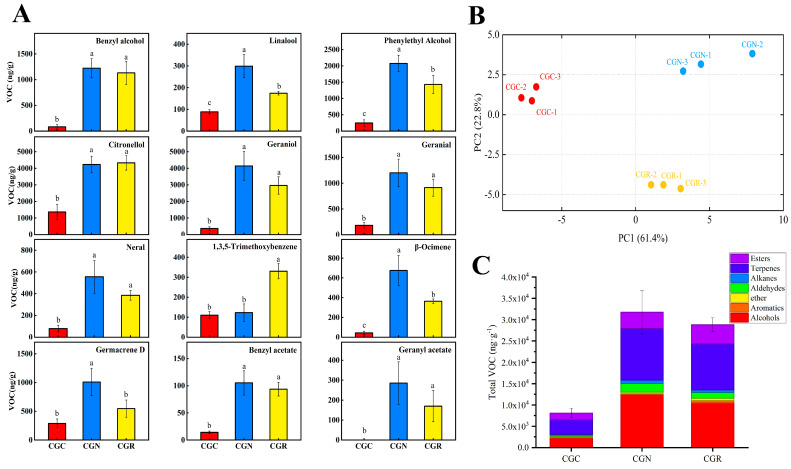
Changes in VOCs in response to different temperatures. CGC, CGN, and CGR refer to storage treatment at 6 °C, 25 °C, and 20 °C. (**A**) Abundance of 12 selected main VOCs composing the floral scent emitted from rose. Significant differences (*p* < 0.05) are denoted by different letters. The error bars indicate the means ± SEs of three biological replicates. (**B**) PCA results of volatiles in roses in response to different temperatures. The volatiles produced by three biological replicates were used as variables. (**C**) Effects of different temperatures on total volatile emissions, which make up seven groups of compounds.

**Figure 3 genes-14-00692-f003:**
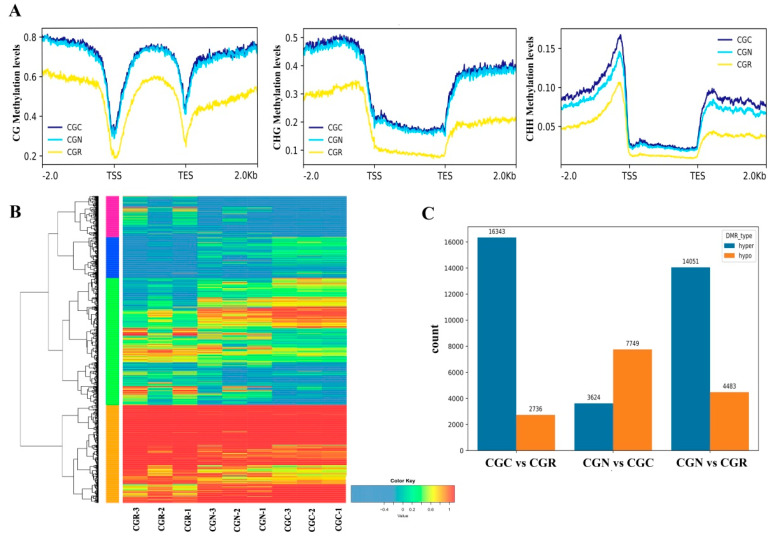
DNA methylation landscape of the rose. (**A**) Genome-wide profiles of CG, CHG, and CHH DNA across the top 7 chromosomes of the rose. The methylation levels of sequences of CG, CHG, and CHH samples subjected to different temperature treatments (6°C, 25°C, and 20 °C) are indicated from the outer to the inner rings. A 1 Mb window was used for calculating the gene/repeat density or methylation level. The red and blue lines inside represent high or low gene densities on the chromosome, respectively. (**B**) Average methylation levels of CG, CHG, and CHH contexts of the roses subjected to different temperature treatments. (**C**) The relative proportion of CG, CHG, and CHH contexts in the total number of methylation sites of rose samples under different temperature treatments.

**Figure 4 genes-14-00692-f004:**
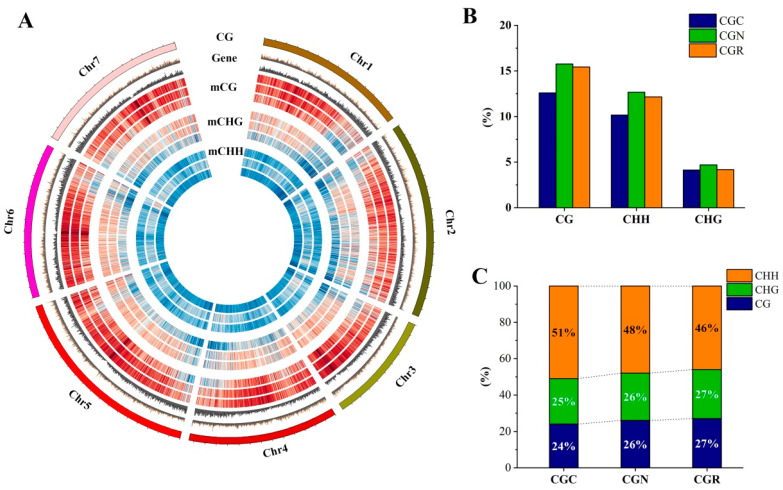
DNA methylation profiles in genes. (**A**) The 2 kb upstream or downstream regions, as well as the gene-body regions, were appropriately divided into 100 equal bins for the calculation of the mean methylation level. TSS: transcription start site; TTS: transcription termination site. (**B**) Differentially expressed genes (DEGs). (**C**) The numbers of DMRs compared with samples of CGC, CGN, and CGR, respectively.

**Figure 5 genes-14-00692-f005:**
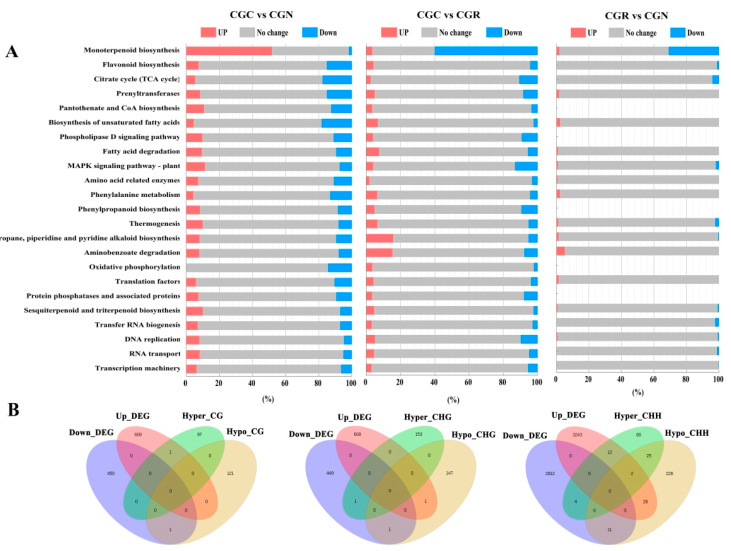
Functional categories of DEGs and Overlap Numbers of DMRs. (**A**) Quantitative patterns of DEGs belonging to specific functional categories defined by MapMan BINs. CGC: plants stored at 6 °C for 24 h. CGR: plants stored at 6 °C for 24 h and then transferred to 20 °C for a 24 h recovery period. CGN: plants grown under normal (25 °C) conditions. The genes in the red BINs are significantly upregulated, the expression of those in the gray BINs is not significantly different, and the genes in the blue BINs are downregulated. (**B**) Numbers of overlapping regions of hypermethylated and hypomethylated-DMRs in the up- or down-regulated DEGs.

**Figure 6 genes-14-00692-f006:**
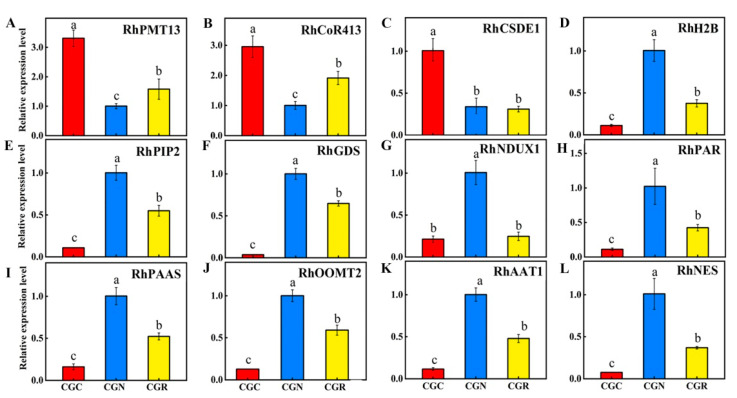
The expression DEGs verified through RT-qPCR. SE values are indicated by vertical bars (n = 3 replicates). Means with different letters (a, b, c) are significantly different (Tukey’s HSD test, P < 0.05). (**A**) *RhPMT13*: Probable methyltransferase PMT13. (**B**) *RhCoR413*: Cold-regulated 413 plasma membrane protein 1. (**C**) *RhCSDE1*: Cold shock domain-containing protein. (**D**) *RhH2B*: Probable histone H2B. (**E**) *RhPIP2*: Aquaporin PIP2-7. (**F**) *RhGDS*: Germacrene D synthase. (**G**) *RhNUDX1*: Nudix hydrolase 1. (**H**) *RhPAP*: Phenylacetaldehyde reductase. (**I**) *RhPAAS*: Phenylacetaldehyde synthase. (**J**) *RhOOMT2*: O-methyltransferase 2. (**K**) *RhAAT1*: Alcohol acetyltransferase. (**L**) *RhNES*: Nerolidol synthase.

**Figure 7 genes-14-00692-f007:**
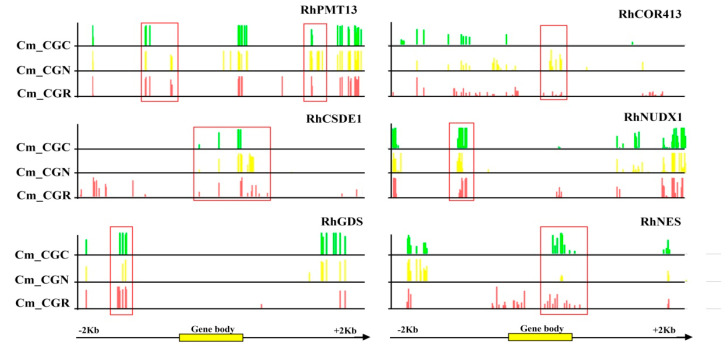
The changes of DNA methylation in DMR-associated genes.

## Data Availability

All raw data has been deposited in the NCBI BioProject database (PRJNA902902). Data Availability Statements in section “MDPI Research Data Policies” at https://www.mdpi.com/ethics.

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
