# Peer review of "Loss of Rose Fragrance under Chilling Stress Is Associated with Changes in DNA Methylation and Volatile Biosynthesis"

_genes, 2023, doi:10.3390/genes14030692_

Round 1

Reviewer 1 Report

Ref: MS entitled “Loss of rose fragrance under chilling is associated with changes in DNA methylation and volatile biosynthesis”. The work is interesting and of reasonable new information for the readers of the journal. I have few comments and suggestions for the improvement.

Please add the word ‘stress’ after chilling in the titles. I mean replace ‘chilling’ with ‘chilling stress’.

The keywords should be different from the title for better indexing of the work.

Please incorporate some related information about the chilling induced loss of fragrance of flowers in the introduction part if available.

Please supplement some maturity related information at which stage the flowers were harvested in section 2.1.

Replace ‘cold’ with ‘chilling’ in the whole manuscript.

How the data were processed/analyzed? I mean statistics if any used. The methodology is usually reproducible with clearly written results.

The discussion may be improved further that how chilling stress regulates floral markers and volatiles biosynthesis by comparing with the recent literature if possible.

Overall, well planned and executed work with new information.

Author Response

Point 1: Please add the word ‘stress’ after chilling in the titles. I mean replace ‘chilling’ with ‘chilling stress’.

Response 1: We have replaced ‘chilling’ with ‘chilling stress’.

Point 2: The keywords should be different from the title for better indexing of the work.

Response 2: We have added one keyword “gene expression”. Our result showed the different expression scent-related genes were associated with the level of DNA methylation.

Point 3: Please incorporate some related information about the chilling induced loss of fragrance of flowers in the introduction part if available.

Response 3: We have incorporated the sentences in the introduction part. “Chilling stress is known to significantly reduce volatile content. Especially 12 volatiles derived from amino acids and lipids were influenced in cold storage.”

Point 4: Please supplement some maturity related information at which stage the flowers were harvested in section 2.1.

Response 4: We have supplied the information in the section 2.1. ‘Crimson Glory’ at the same floral developmental stages 4 (flowers start to open) were stored.

Point 5: Replace ‘cold’ with ‘chilling’ in the whole manuscript.

Response 5: We have replaced ‘cold’ with ‘chilling’ in the whole manuscript.

Point 6: How the data were processed/analyzed? I mean statistics if any used. The methodology is usually reproducible with clearly written results.

Response 6: In this study, volatiles were processed by principal component analysis (PCA). We have added the methods in the MS. In addition, We have cited the references for other data. For example, the differentially methylated genomic regions of R. ‘Crimson Glory’ in response to chilling stress were identified via the METHYLKIT package [30]. Expression fold-changes were calculated by means of the cycle threshold (CT) method [33]

Point 7: The discussion may be improved further that how chilling stress regulates floral markers and volatiles biosynthesis by comparing with the recent literature if possible.

Response 7: In the discussion part, we have stated the content and added the references in the MS. Chilling stress regulates floral markers and volatiles biosynthesis, especially RhNUDX1, RhGDS, and RhPAR through changes in DNA methylation. 

Reviewer 2 Report

Dear Editor,

Thank you for inviting me to review the paper titled: Loss of rose fragrance under chilling is associated with changes in DNA methylation and volatile biosynthesis

The study title is interesting, in contrast, the author didn't describe bioinformatics analysis well.

The manuscripts were not organized very well in the bioinformatics analysis. Moreover, the project number reported in the manuscript is not available in NCBI, and the methodology of analysis does not explain clearly, the supplementary files are not available. The range of foldchange is not available.

Why didn't preprocess the data?

How to integrate data?

It is recommended to add some literature about cold. For example the authors may address to the following papers:

https://doi.org/10.17660/ActaHortic.2010.861.35

https://doi.org/10.1016/j.scienta.2017.06.058

The DEG was calculated by which package?

The enrichments were missed in the analysis.

Author Response

Point 1: Thank you for inviting me to review the paper titled: Loss of rose fragrance under chilling is associated with changes in DNA methylation and volatile biosynthesis

The study title is interesting, in contrast, the author didn't describe bioinformatics analysis well. The manuscripts were not organized very well in the bioinformatics analysis. Moreover, the project number reported in the manuscript is not available in NCBI, and the methodology of analysis does not explain clearly, the supplementary files are not available. The range of fold change is not available.

Response 1: We have deposited the raw data of WGBS-seq and RNA-seq from R. ‘Crimsin Glory’ in NCBI. The accession number PRJNA902902. We have received the information from NCBI as follows:

Accession: SAMN31956623, Sample Name: CGM-N, CGM-R, CGM-C,  Organism: Rosa, Tax ID: 74649, Cultivar: Crimsin Glory, BioProject: PRJNA902902. The data will be available when the manuscript is public.

In addition, the supplementary files are made from five excel tables. I believe these files are available.

Point 2: Why didn't preprocess the data? How to integrate data?

Response 2: We have deposited the raw data to NCBI.

In result part: We have integrated data of WGBS-seq and RNA-seq in the MS:

To explore the correction between DNA methylation and different gene expression in rose, we performed the transcriptome analysis of the identical samples with methylation analysis and compared the DEGs and DMRs in rose in response to chilling stress.

Point 3: It is recommended to add some literature about cold. For example the authors may address to the following papers:

https://doi.org/10.17660/ActaHortic.2010.861.35

https://doi.org/10.1016/j.scienta.2017.06.058

Response 3: We have added three references about cold in the MS, including two papers mentioned. Thank you very much.

Point 4: The DEG was calculated by which package? The enrichments were missed in the analysis.

Response 4: The DEG was calculated according to the statistical R package DEGseq using MA-plot-based method with random sampling model. In the study, we haven’t showed the enrichments analysis in the MS. And DEGs were classified into functional categories defined by MapMan BINs (mapman.gabipd.org).
